# PUFA synthase-independent DHA synthesis pathway in *Parietichytrium* sp. and its modification to produce EPA and n-3DPA

Yohei Ishibashi[1,8], Hatsumi Goda[1,8], Rie Hamaguchi[1,8], Keishi Sakaguchi[1,8], Takayoshi Sekiguchi[2,8], Yuko Ishiwata[2], Yuji Okita[2], Seiya Mochinaga[1], Shingo Ikeuchi[1], Takahiro Mizobuchi[1], Yoshitake Takao[3], Kazuki Mori[1], Kosuke Tashiro[1], Nozomu Okino[1], Daiske Honda[4,5], Masahiro Hayashi[6] & Makoto Ito [1,7 ✉]

The demand for n-3 long-chain polyunsaturated fatty acids (n-3LC-PUFAs), such as docosahexaenoic acid (DHA) and eicosapentaenoic acid (EPA), will exceed their supply in the near future, and a sustainable source of n-3LC-PUFAs is needed. Thraustochytrids are marine protists characterized by anaerobic biosynthesis of DHA via polyunsaturated fatty acid synthase (PUFA-S). Analysis of a homemade draft genome database suggested that *Parietichytrium* sp. lacks PUFA-S but possesses all fatty acid elongase (ELO) and desaturase (DES) genes required for DHA synthesis. The reverse genetic approach and a tracing experiment using stable isotope-labeled fatty acids revealed that the ELO/DES pathway is the only DHA synthesis pathway in *Parietichytrium* sp. Disruption of the C20 fatty acid ELO (C20ELO) and $\Delta 4$ fatty acid DES ($\Delta 4$DES) genes with expression of $\omega 3$ fatty acid DES in this thraustochytrid allowed the production of EPA and n-3docosapentaenoic acid (n-3DPA), respectively, at the highest level among known microbial sources using fed-batch culture.

[1] Department of Bioscience and Biotechnology, Faculty of Agriculture, Kyushu University, Fukuoka 819-0395, Japan. [2] Central Research Laboratory, Nippon Suisan Kaisha, Ltd., Tokyo 192-0991, Japan. [3] Department of Marine Science and Technology, Faculty of Marine Science and Technology, Fukui Prefecture University, Fukui 917-0003, Japan. [4] Department of Biology, Faculty of Science and Engineering, Konan University, Hyogo 658-8501, Japan. [5] Institute for Integrative Neurobiology, Konan University, Hyogo 658-8501, Japan. [6] Department of Marine Biology and Environmental Sciences, Faculty of Agriculture, University of Miyazaki, Miyazaki 889-2192, Japan. [7] Innovative Bio-architecture Center, Faculty of Agriculture, Kyushu University, Fukuoka 819-0395, Japan. [8] These authors contributed equally: Yohei Ishibashi, Hatsumi Goda, Rie Hamaguchi, Keishi Sakaguchi, Takayoshi Sekiguchi. ✉email: makotoi@agr.kyushu-u.ac.jp

Docosahexaenoic acid (DHA, C22:6n-3) is an n-3 long-chain polyunsaturated fatty acid (n-3LC-PUFA) that exists in organisms ranging from certain microbes to mammals but not in higher plants[1–3]. Marine microbes, such as marine bacteria and protists (thraustochytrids), synthesize DHA from acetyl-CoA and malonyl-CoA via polyketide-synthase like protein named PUFA synthase (PUFA-S)[4–6]. Among thraustochytrids, *Schizochytrium* sp.[4] and *Aurantiochytrium limacinum*[5] are known to synthesize DHA via PUFA-S, but several genes related to the fatty acid elongase (ELO)/desaturase (DES) pathway are present in the genome of *A. limacinum*[7], and further analysis is therefore needed. *Thraustochytrium* sp. and *T. aureum* produces DHA via two distinct pathways, namely, the PUFA-S and ELO/DES pathways[8,9]. Although the contribution of the ELO/DES pathway to DHA production is considered to be relatively small in both *Thraustochytrium*[10,11], quantitative data are presently limited.

n-3LC-PUFAs, such as DHA and eicosapentaenoic acid (EPA, C20:5n-3), are beneficial both nutritionally and pharmacologically and are used as supplements and medicines[12–14]. These n-3LC-PUFAs are industrially produced from oily fish, although the primary producers of n-3LC-PUFAs are mainly microalgae and thraustochytrids. However, there is concern that the supply of n-3LC-PUFAs will be insufficient in the future due to a decrease in fish resources resulting from climate change and overfishing[15]. The n-3LC-PUFA market is expected to be worth 4300 million USD by 2019[16], and an increase in water temperature due to global warming may result in a 10–58% loss in the globally available DHA by 2100[17]. Thus, sustainable production of n-3LC-PUFAs using genetically engineered oilseed crops[2,18] and microbial fermentation[2,19,20] is needed as an alternative to production from fish oil. Among unicellular microbes, thraustochytrids have been used industrially because they can reach high cell densities with a high DHA yield[21].

In addition to DHA, EPA and arachidonic acid (ARA, C20:4n-6) are other LC-PUFAs that have drawn major interest for their role in human health and nutrition[22,23]. The purified ethyl ester of EPA (commercial names Epadel and Vascepa) has been approved as a therapeutic agent for obstructive arteriosclerosis and hypertriglyceridemia[24,25]. These medicines reduce the serum triacylglycerol (TAG) level[26]. EPA and DHA may have different physiological and pharmacological functions in humans[27]. Although data remain limited, n-3 docosapentaenoic acid (n-3DPA, C22:5n-3), an intermediate metabolite of DHA in the ELO/DES pathway, is also expected to have health benefits[28–30].

We report here that *Parietichytrium* sp. does not have PUFA-S and synthesizes DHA exclusively through the ELO/DES pathway. To the best of our knowledge, *Parietichytrium* sp. is the first example of a thraustochytrid that synthesizes a large amount of DHA through a mechanism other than the PUFA-S pathway. We have achieved the production of EPA and n-3DPA at the highest level among known microbial sources through the genetic manipulation of PUFA-S-independent DHA synthesis pathway in *Parietichytrium* sp.

## Results

*Schizochytrium* sp. ATCC20888[4,31] and *A. limacinum* synthesize DHA anaerobically via PUFA-S[5], whereas *Thraustochytrium* sp. ATCC26185[8] and *T. aureum* ATCC34304[9] produces DHA via both the anaerobic PUFA-S and aerobic ELO/DES pathways. These two pathways in thraustochytrids are illustrated in Fig. 1a. Differences in the DHA synthesis systems are reflected in their LC-PUFA composition (Fig. 1b). That of *A. limacinum* is relatively simple and consists mainly of DHA and n-6 docosapentaenoic acid (n-6DPA, C22:5n-6), whereas that of *T.*

*aureum* is slightly complex, and considerable amounts of EPA and ARA are included, in addition to DHA and n-6DPA. The LC-PUFA composition of *Parietichytrium* sp. showed greater diversity, and the levels of n-3DPA and docosatetraenoic acid (DTA, C22:4n-6) increased (Fig. 1b). The total fatty acid compositions of three thraustochytrids are also shown in Table 1, which indicates high level of oleic acid (OA, C18:1n-9) in *Parietichytrium* sp. The difference in fatty acid composition, especially LC-PUFA (Fig. 1b), may indicate that *Parietichytrium* sp. has a different LC-PUFA biosynthetic system from those of *A. limacinum* and *T. aureum*.

Thus, we examined draft genome databases of *A. limacinum* (provided by the U.S. Department of Energy, Joint Genome Institute, USA), *T. aureum* (this study) and *Parietichytrium* sp. (this study) (Table 2) and found that the three thraustochytrid species share the fatty acid synthase (FAS) gene that functions in the synthesis of palmitic acid (C16:0, PA). The PUFA-S gene was found in *A. limacinum* and *T. aureum* but not in *Parietichytrium* sp., suggesting that *Parietichytrium* sp. does not synthesize DHA via PUFA-S. On the other hand, the Δ4DES gene, which is responsible for the conversion of n-3DPA to DHA and is essential for DHA synthesis in the ELO/DES pathway[32], was found in *T. aureum* and *Parietichytrium* sp. but not in *A. limacinum*. These results are basically consistent with previous reports that *A. limacinum* can synthesize DHA via the PUFA-S pathway, and *Thraustochytrium* (*Thraustochytrium* sp. and *T. aureum*) can synthesize DHA via both the PUFA-S and ELO/DES pathways[5,9]. In addition, the present genome analysis strongly suggested that *Parietichytrium* sp. synthesizes DHA only through a PUFA-S-independent pathway.

To disclose the PUFA-S independent pathway for DHA synthesis, we cloned all ELO and DES genes constituting ELO/DES pathway of *Parietichytrium* sp. (Supplementary Fig. S1). This pathway consists of three ELOs (tentatively designated ELO-1, ELO-2, and ELO-3) and six DESs (DES-1, DES-2, DES-3, DES-4, DES-5, and DES-6). We analyzed the substrate specificity of each enzyme by analysis of fatty acid profiles in budding yeasts after expression of each ELO or DES gene. As a result, we found that DES-1, DES-2, DES-3, DES-4, DES-5, and DES-6 were Δ9DES, Δ12DES (ω6DES), Δ6DES, Δ5DES, Δ4DES, and Δ17/19DES (ω3DES), respectively, and ELO-1, ELO-2, and ELO-3 were C16ELO, C18/C20ELO (Δ6ELO), and C20ELO (Δ5ELO), respectively (Supplementary Fig. S1a, b). All the ELO/DES genes examined were actually expressed in *Parietichytrium* sp.; however, their expression levels varied; the expression of DES-1, DES-2, and DES-3 was high, that of ELO-1, ELO-2, ELO-3, and DES-4 was moderate, and that of DES-5 and DES-6 was low (Supplementary Fig. S1c).

To address whether *Parietichytrium* sp. synthesizes DHA only through the ELO/DES pathway, we generated a Δ4DES-deficient mutant (Δ4DES KO) of *Parietichytrium* sp. and compared the fatty acid composition between the mutant and wildtype (WT). As shown in Fig. 1c, the Δ4DES KO of *Parietichytrium* sp. completely lacked DHA and n-6DPA, whereas the amount of n-3DPA and DTA (C22:4n-6), which are precursors of DHA and n-6DPA in the ELO/DES pathway (Fig. 1a), respectively, markedly increased. The profiles of gas chromatography (GC) of *Parietichytrium* sp. WT and its Δ4DES KO mutant is also shown in Supplementary Fig. S2. These results indicate that DHA and n-6DPA are generated through only the ELO/DES pathway in *Parietichytrium* sp., in which Δ4DES is the sole enzyme that converts n-3DPA to DHA and DTA to n-6DPA. In contrast to *Parietichytrium* sp., DHA and n-6DPA were still present in the Δ4DES KO of *T. aureum*, and these LC-PUFAs were lost when both the PUFA-S and Δ4DES genes were disrupted (Supplementary Fig. S3a).

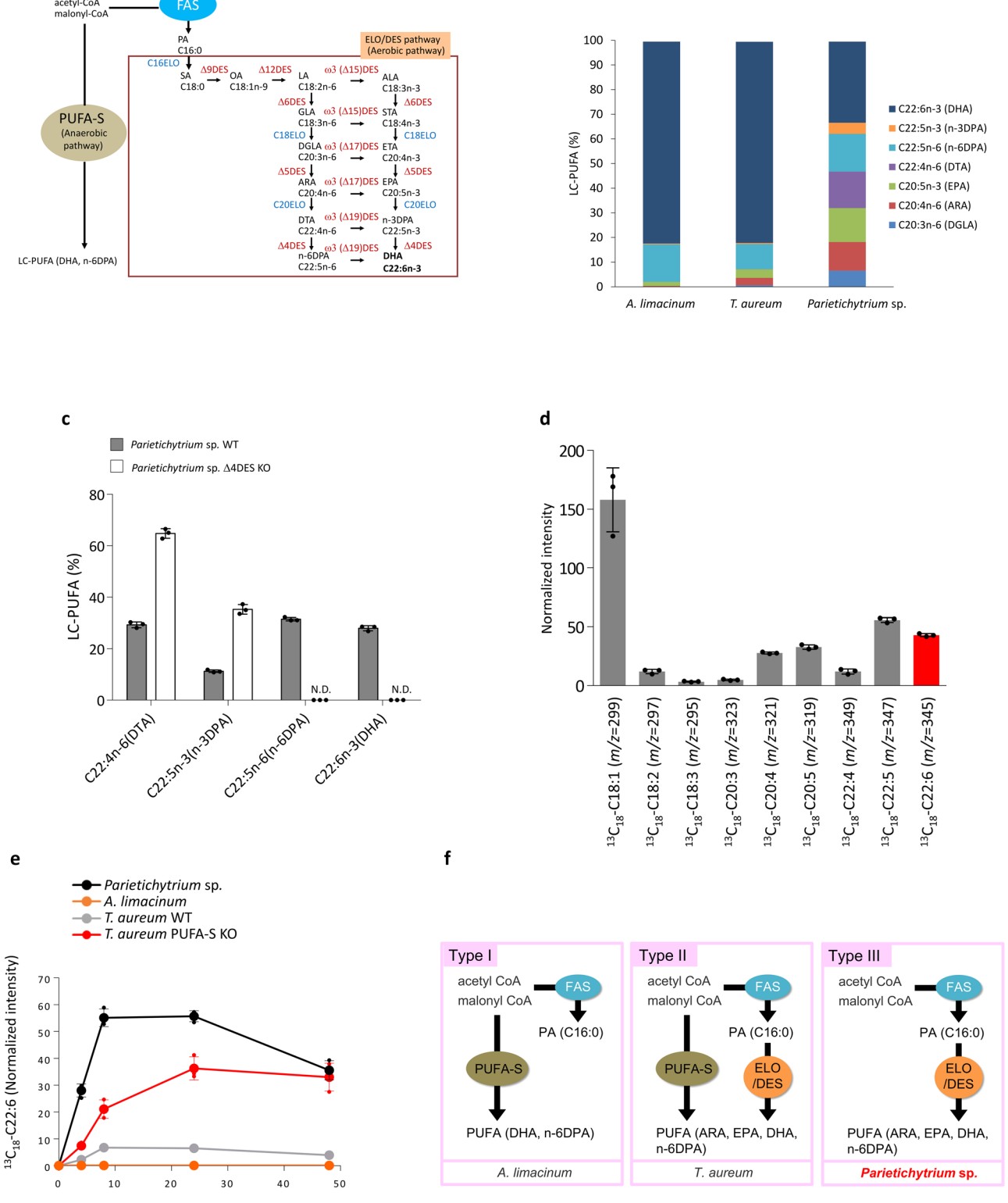

To confirm that *Parietichytrium* sp. produces DHA through the ELO/DES pathway, we traced the DHA synthesis of *Parietichytrium* sp. using $^{13}$C-labeled OA ($^{13}C_{18}$-C18:1) as a precursor. *Parietichytrium* sp. was cultured in GY medium containing $^{13}C_{18}$-C18:1, and total fatty acids were extracted from the cells after cultivation for 24 h. The fatty acid fraction was then analyzed by LC-ESI MS/MS as described in the Methods, and

multiple reaction monitoring (MRM) conditions for $^{13}$C-labeled fatty acids are described in Supplementary data set 1. We were able to detect the precursor $^{13}C_{18}$-C18:1 and end product $^{13}C_{18}$-labeled DHA ($^{13}C_{18}$-C22:6) as well as intermediates during DHA synthesis by the ELO/DES pathway (Fig. 1d). The production of $^{13}C_{18}$-C22:6 in *Parietichytrium* sp. reached a plateau at 9–24 h and began to decline after 24 h (Fig. 1e). The time-course for the

**Fig. 1 Identification and classification of DHA synthesis systems of thraustochytrids. a** Two pathways for the synthesis of n-6DPA and DHA in thraustochytrids. One occurs via PUFA-S, directly synthesizing n-6DPA or DHA from acetyl-CoA and malonyl-CoA in a polyketide synthase-like manner. The other is the ELO/DES pathway, in which PA is converted to DHA through six desaturation and three elongation steps. **b** LC-PUFA compositions of three different genera of thraustochytrids (*A. limacinum*, *T. aureum*, and *Parietichytrium* sp.SEK358). Each strain was cultured in 20 mL of GY medium at 28 °C for 3 days with shaking at 120 rpm. LC-PUFAs were determined by GC. **c** LC-PUFA profiles of WT and Δ4DES KO of *Parietichytrium* sp. SEK358. The data shown are the mean ± SD. ($n = 3$). The $n$ values are numbers of replicates. **d** Generation of $^{13}C_{18}$-labeled 22:6 (DHA) and its intermediates from the precursor $^{13}C_{18}$-C18:1 in *Parietichytrium* sp. Cells of *Parietichytrium* sp. SEK358 were incubated at 25 °C with 0.25 mM $^{13}C_{18}$-C18:1 for 24 h. The $^{13}C_{18}$-labeled fatty acids were obtained from total lipid fractions after alkaline treatment and were determined by MRM analysis using LC-ESI MS/MS, as shown in Supplementary data set 1. Peak intensities of $^{13}C_{18}$-labeled fatty acids were normalized to an internal standard (C12:0) (normalized intensity). The data shown are the mean ± SD. ($n = 3$). The $n$ values are numbers of replicates. **e** Time course of the generation of $^{13}C_{18}$-C22:6 generated from $^{13}C_{18}$-C18:1 in *Parietichytrium* sp., *A. limacinum*, *T. aureum* WT, and *T. aureum* PUFA-S KO. Cells were incubated with 0.25 mM $^{13}C_{18}$-C18:1 and collected at the indicated time points. The data shown are the mean ± SD. ($n = 3$). The $n$ values are numbers of replicates. **f** Schematic diagrams for three types of DHA synthesis systems in thraustochytrids. DHA is synthesized only by PUFA-S in type I (e.g., *A. limacinum*), by the ELO/DES pathway in type III (e.g., *Parietichytrium* sp.), or by both the PUFA-S and ELO/DES pathways in type II (e.g., *T. aureum*).

---

**Table 1 Total fatty acid compositions of *A. limacinum*, *T. aureum*, and *Parietichytrium* sp.**

|  | *A. limacinum* | *T. aureum* | *Parietichytrium* sp. |
|---|---|---|---|
| C14:0 (MA) | 2.2 ± 0.16 | 1.7 ± 0.09 | 4.6 ± 0.54 |
| C16:0 (PA) | 38.4 ± 1.15 | 22.6 ± 0.37 | 23.6 ± 0.26 |
| C18:0 (SA) | 1.3 ± 0.03 | 18.7 ± 0.18 | 22.6 ± 2.23 |
| C18:1n-9 (OA) | n.d. | 2.2 ± 0.2 | 27.0 ± 1.56 |
| C18:2n-6 (LA) | n.d. | 0.4 ± 0.01 | 3.1 ± 0.14 |
| C18:3n-6 (GLA) | n.d. | 0.1 ± 0.01 | 0.3 ± 0.02 |
| C18:3n-3 (ALA) | 0.1 ± 0.06 | n.d. | n.d. |
| C18:4n-3 (STA) | 0.1 ± 0.06 | 0.2 ± 0.03 | n.d. |
| C20:2n-6 (EDA) | n.d. | 0.2 ± 0.05 | 0.4 ± 0.03 |
| C20:3n-6 (DGLA) | n.d. | 0.3 ± 0.01 | 1.0 ± 0.17 |
| C20:4n-6 (ARA) | 0.1 ± 0.02 | 1.6 ± 0.1 | 1.8 ± 0.27 |
| C20:3n-3 (ETrA) | n.d. | n.d. | n.d. |
| C20:4n-3 (ETA) | 0.3 ± 0.02 | 0.3 ± 0.01 | 0.1 ± 0.01 |
| C20:5n-3 (EPA) | 0.9 ± 0.04 | 1.8 ± 0.12 | 2.1 ± 0.17 |
| C22:4n-6 (DTA) | n.d. | n.d. | 2.3 ± 0.44 |
| C22:5n-6 (n-6DPA) | 8.6 ± 0.17 | 5.3 ± 0.09 | 2.4 ± 0.2 |
| C22:5n-3 (n-3DPA) | 0.2 ± 0.01 | 0.2 ± 0.02 | 0.7 ± 0.07 |
| C22:6n-3 (DHA) | 45.9 ± 0.9 | 42.2 ± 0.59 | 5.1 ± 0.3 |
| Others | 2.0 ± 0.24 | 2.2 ± 0.31 | 2.9 ± 0.14 |

Total fatty acid compositions of three different genera of thraustochytrids. Each strain was cultured in 20 mL of GY medium at 28 °C for 3 days with shaking at 120 rpm. Fatty acid compositions were determined by GC analysis. n.d., not detected.

---

**Table 2 Genes responsible for the synthesis of DHA found in the draft genomes of *A. limacinum*, *T. aureum*, and *Parietichytrium* sp.**

|  |  | *A. limacinum* Type I | *T. aureum* Type II | *Parietichytrium* sp. Type III |
|---|---|---|---|---|
| FAS | Query seq. | *Schizochytrium* type I FAS | *Schizochytrium* type I FAS | *Schizochytrium* type I FAS |
|  | *E*-value | 0[a] | 2e−178[a] | 0[a] |
| PUFA-S | Query seq. | *Schizochytrium* PUFA-S A/B/C | *Schizochytrium* PUFA-S A/B/C | *Schizochytrium* PUFA-S A/B/C |
|  | *E*-value | 0[a]/0[a]/0[a] | 0[a]/0[a]/0[a] | 5e−43[b]/2e−25[c]/not found[d] |
| Δ4DES | Query seq. | *T. aureum* Δ4DES | *T. aureum* Δ4DES | *T. aureum* Δ4DES |
|  | *E*-value | not found[d] | 0[a] | 0[a] |

Draft genome databases of *A. limacinum* ATCC MYA-1381 (provided by the U.S. Department of Energy, Joint Genome Institute, USA), *T. aureum* ATCC34304 (this study), and *Parietichytrium* sp. I65-24A (this study) were searched using amino acid sequences of type I FAS (EF015632) from *Schizochytrium* sp. ATCC 20888, PUFA-S subunit A (AF378327), subunit B (AF378328), subunit C (AF378329) from *Schizochytrium* sp. ATCC 20888, and Δ4DES from *T. aureum* ATCC 34304 (AF391543). The E-value of each gene is described in the table.
[a]A corresponding sequence was found in each database.
[b,c]A homologous sequence was found in the *Parietichytrium* database, but the highest homologous sequence was found in [b]*Marinagarivorans algicola* as "hybrid nonribosomal peptide synthetase/type I polyketide synthase" (*E*-value = 0.0) or [c]*Arabidopsis thaliana* as "3-oxoacyl carrier protein synthase" (*E*-value = 1e−116).
[d]Homologous sequences (*E*-value < 10) were not found in each database.

generation of intermediates of DHA synthesis in *Parietichytrium* sp. is shown in Supplementary Fig. S4. In *T. aureum* WT, the production of $^{13}C_{18}$-C22:6 was much lower than that of *Parietichytrium* sp.; however, it significantly increased when the PUFA-S gene was disrupted (*T. aureum* PUFA-S KO, Fig. 1e), suggesting that the ELO/DES pathway is a supplementary or auxiliary system for the PUFA-S pathway in DHA synthesis in *T. aureum*. No $^{13}C_{18}$-C22:6 was produced in *A. limacinum* under the same conditions, indicating that DHA is not produced by the ELO/DES pathway in *A. limacinum* (Fig. 1e). It was reported, however, that several genes related to the ELO/DES pathway are present in the genome of *A. limacinum*[7], and thus, a tracing experiment using stable isotope-labeled PA (d31–C16:0) was conducted to determine which step of the ELO/DES pathway in *A. limacinum* does not function in vivo. We found that d31–C18:0 and d29–C18:1 were generated in *Parietichytrium* sp. and *T. aureum* (WT) when d31–C16:0 was added to each culture; however, in *A. limacinum* d29–C18:1 could not be detected at all, although a small amount of d31–C16:0 was converted to d31–C18:0 (Supplementary Fig. S3b, c). This result indicates that the machinery for the conversion of C16:0 to C18:1 is not fully functional in *A. limacinum*, and in particular, Δ9DES, which is responsible for the conversion of C18:0 to C18:1, does not function in vivo. The fact that the C16ELO homolog exists in the *A. limacinum* draft genome database but the Δ9DES homolog does not, which is consistent with the results obtained in this study. Thus, we conclude that the ELO/DES pathway does not function in vivo due at least to the lack of Δ9DES function and that DHA is exclusively generated through the PUFA-S pathway in *A. limacinum*.

These results indicate that thraustochytrids can be classified into three types according to their DHA synthesis system: type I, which synthesizes DHA via only the PUFA-S pathway, such as *A. limacinum*; type II, which has both PUFA-S and ELO/DES pathways, such as *T. aureum*; and type III, which synthesizes DHA via only the ELO/DES pathway, such as *Parietichytrium* sp. (Fig. 1f).

The major LC-PUFAs of *Parietichytrium* sp. were DHA and n-6DPA, and the levels of EPA and ARA were relatively low (Fig. 1b and Table 1). Disruption of the gene encoding ELO-3 (C20ELO) in *Parietichytrium* sp. resulted in a reduction in DHA and n-6DPA levels and a marked increase in EPA and ARA levels (Fig. 2a). The GC profiles are also shown in supplementary Fig. S5. A small amount of DHA and n-6DPA remained in the ELO-3 gene-deficient mutant (C20ELO KO), probably because ELO-2 (C18/C20ELO) partially compensated for the defect of ELO-3 (C20ELO) in C20ELO KO. The expression level of DES-6 (ω3DES) in *Parietichytrium* sp. was markedly low (Supplementary Fig. S1c); therefore, we examined whether heterologous expression of ω3DES could increase the amount of EPA in C20ELO KO cells. Prior to this experiment, we searched for suitable ω3DES genes and potent promoters, by which ARA is expected to be efficiently converted to EPA in vivo. Among the three ω3DESs tested, ω3DES from *Saprolegnia diclina*[33] most efficiently converted ARA to EPA (Supplementary Fig. S6a). We also found that the Smp1 promoter (Smp1P) from *Sicyoidochytrium minutum* DNA virus (SmDNAV)[34] exhibited several times higher expression efficiency than the conventional ubiquitin promoter[35] when the neomycin resistance gene (NeoR) was used as a marker gene (Supplementary Fig. S6b, c). Thus, we expressed the *S. diclina* ω3DES gene in C20ELO KO cells using an over-expression construct driven by the Smp1 promoter (Supplementary Fig. S6d). As a result, the EPA level in the mutant (C20ELO KO/ω3DES OE) increased 6-fold compared with that in the WT (Fig. 2a). The growth of two mutants (C20ELO KO and C20ELO KO/ω3DES OE) was compared with that of the WT in

flask culture. The growth in log phase of the three strains was almost the same based on either total biomass (dry cell weight, DCW) or glucose consumption; however, the growth of the two mutants rapidly declined compared with that of the WT after day 3 under the culture conditions used (Fig. 2b). Similar to the growth, the total fatty acid (TFA) content of these mutants also decreased much faster than that of the WT after day 3 (Fig. 2c). Figure 2d and e show the time-courses of the production of ARA and EPA, respectively, in the WT and two mutants. In C20ELO KO, the production of ARA was much higher than that in the WT, but the production of EPA was lower than that of ARA. However, the production of EPA exceeded that of ARA in C20ELO KO/ω3DES OE throughout cultivation and reached 350 mg/L at day 3, indicating that *S. diclina* ω3DES efficiently converted ARA to EPA in this mutant. The increase in ARA and EPA in C20ELO KO and the conversion of ARA to EPA in C20ELO KO/ω3DES OE were observed in both the phospholipid (phosphatidylcholine, PC) and neutral lipid (TAG) fractions by LC-ESI MS/MS analysis (Supplementary Fig. S7).

Disruption of the gene encoding DES-5 (Δ4DES) in *Parietichytrium* sp. resulted in marked accumulation of n-3DPA and DTA, with complete loss of DHA and n-6DPA (Fig. 2f). Similar to C20ELO KO/ω3DES OE, the expression of *S. diclina* ω3DES in Δ4DES KO facilitated the conversion of n-6LC-PUFA (DTA) to n-3LC-PUFA (n-3DPA), resulting in a 11-fold increase in n-3DPA production compared with that in the WT (Fig. 2f). Unlike C20ELO disruption, Δ4DES disruption did not cause notable growth inhibition of the mutant (Fig. 2g). After day 3, a decrease in TFA was observed in the two mutants (Δ4DES KO and Δ4DES KO/ω3DES OE), but the extent of this decrease was similar to that in the WT (Fig. 2h). Figure 2i and j show the production of DTA and n-3DPA, respectively, in the flask culture. The production of n-3DPA was always higher than that of DTA in Δ4DES KO/ω3DES OE mutant during the course of cultivation (Fig. 2j) and reached 300 mg/L at days 3 and 4. LC-ESI MS/MS confirmed that the n-3DPA level of PC and TAG increased in both mutants (Supplementary Fig. S8).

The amounts of TFA and biomass (DCW) decreased from day 3 when the glucose in the medium was exhausted in flask culture (Fig. 2b, g), suggesting that the decrease in TFA and biomass was due to the depletion of glucose in the medium. In addition, the air supply may be insufficient in flask culture, which can inhibit the production of LC-PUFAs in *Parietichytrium* sp. because LC-PUFAs are exclusively produced via the ELO/DES pathway, in which DES requires molecular oxygen for its reaction. Thus, we examined the effects of air supply on the production of LC-PUFAs using three different concentrations of dissolved oxygen (DO) in flask culture (Supplementary Fig. S10). As a result, an increase in the oxygen supply significantly increased the proportion of LC-PUFAs in TFA of *Parietichytrium* sp. but not in TFA of *A. limacinum* (Fig. 3a), which synthesizes LC-PUFAs only via an anaerobic PUFA-S pathway. These results suggest that LC-PUFA production in *Parietichytrium* sp. can be improved by increasing the glucose and oxygen supply in fed-batch culture.

We examined the conditions for fed-batch culture of *Parietichytrium* sp. mutant strains. A good yield of EPA production in a 1 L fed-batch culture was obtained using a medium containing 3% glucose and 2% yeast extract in 50% artificial sea water supplemented with a vitamin mixture and an element solution. Of note, the addition of inorganic nitrogen compounds such as monosodium glutamate, ammonium sulfate, and potassium phosphates as a feed solution was effective for increasing DCW and TFA, leading to an increase of EPA production (Supplementary Fig. S11). Under these conditions, the C20ELO KO/ω3DES OE and Δ4DES KO/ω3DES OE mutants showed increased proportions of EPA and n-3DPA, respectively, in LC-

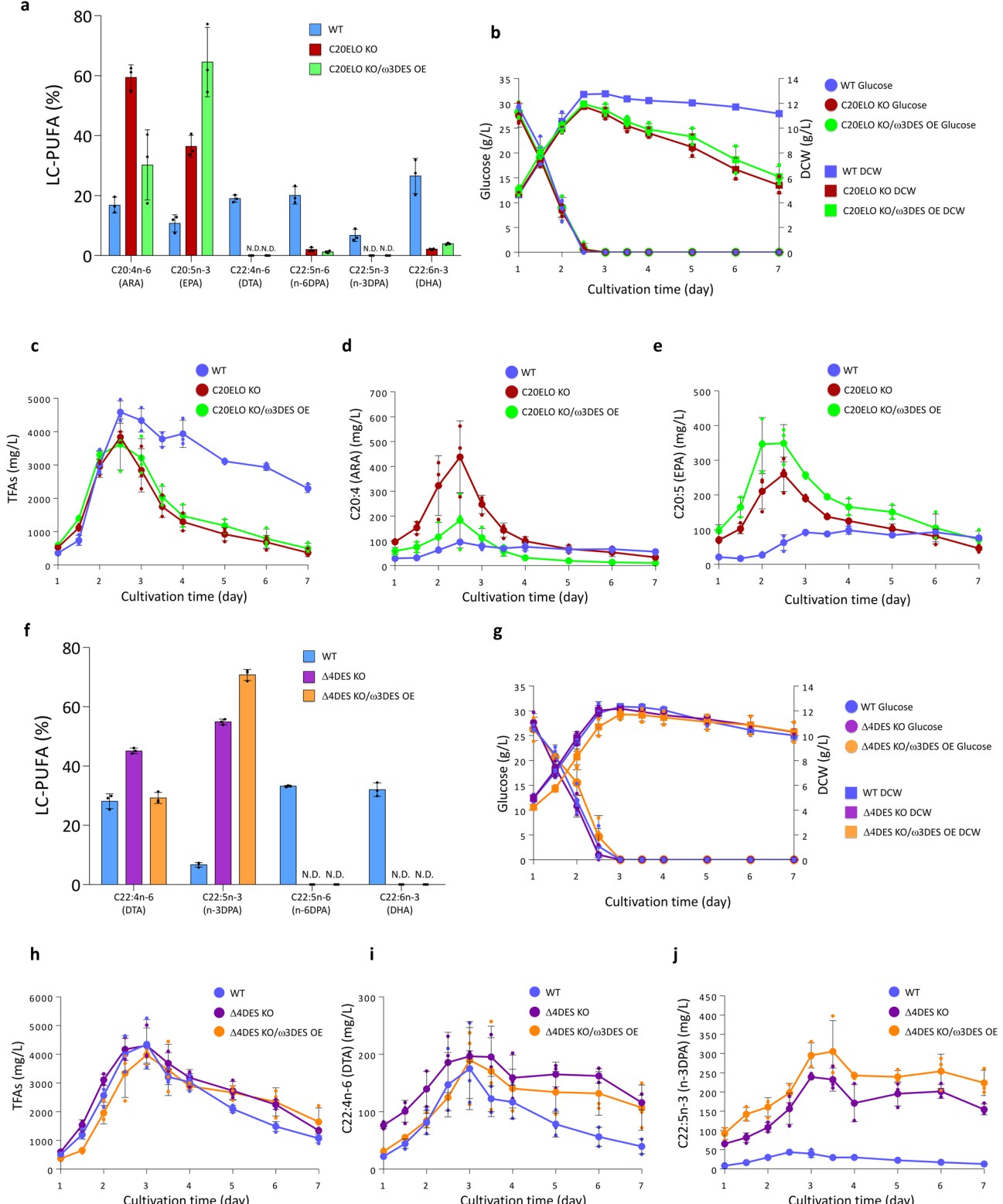

**Fig. 2 Production of EPA and n-3DPA by genetic manipulation of *Parietichytrium* sp. a** LC-PUFA profiles of WT, C20ELO KO, and C20ELO KO/ω3DES OE of *Parietichytrium* sp. in flask culture. Fatty acids were extracted from the 2-day cultured *Parietichytrium* sp. and used for GC analysis. **b** Growth characteristics of each strain. Glucose consumption and DCW were measured. **c** Time course of TFA production in each strain. **d** Time course of the production of C20:4n-6 (ARA) in each strain. **e** Time course of the production of C20:5 (EPA) in each strain. **f** LC-PUFA profiles of WT, Δ4DES KO, and Δ4DES KO/ω3DES OE of *Parietichytrium* sp. in flask culture. Fatty acids were extracted from the 3-day cultured *Parietichytrium* sp. and used for GC analysis. **g** Growth characteristics (glucose consumption and DCW) of each strain. **h** Time course of TFA production in each strain. **i** Time course of the production of C22:4n-6 (DTA) in each strain. **j** Time course of the production of C22:5n-3 (n-3DPA) in each strain. All data shown are the mean ± SD. (*n* = 3). The *n* values are numbers of replicates. WT, C20ELO KO, C20ELO KO/ω3DES OE, Δ4DES KO, and Δ4DES KO/ω3DES OE strains used in flask culture were derived from *Parietichytrium* sp. SEK358.

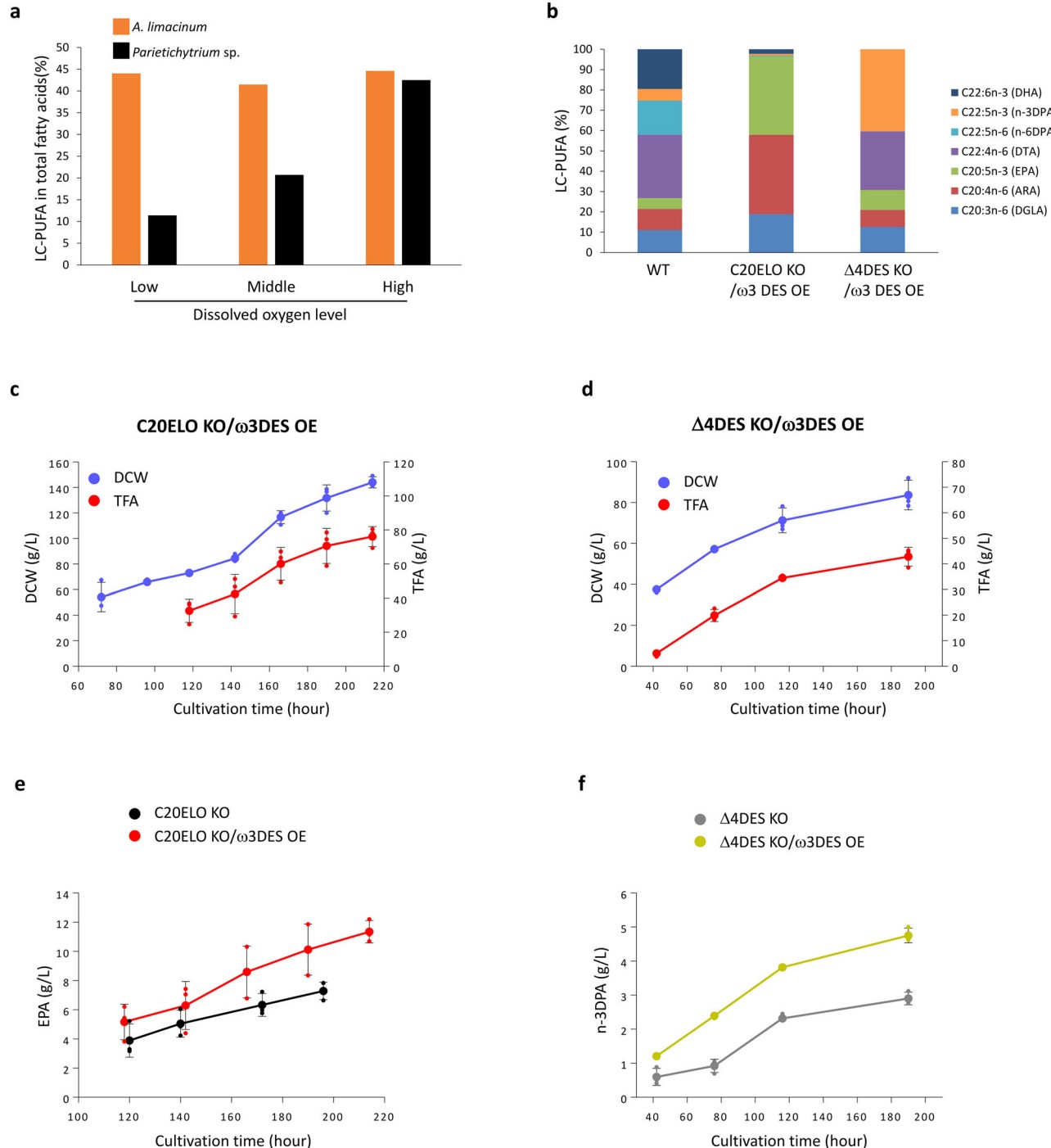

**Fig. 3 Production of EPA and n-3DPA using *Parietichytrium* sp. mutant strains by fed-batch culture. a** LC-PUFA levels of *Parietichytrium* sp. and *A. limacinum* cultured in the GY medium with different air supply. **b** LC-PUFA profiles of WT, C20ELO KO/ω3DES OE, and Δ4DES KO/ω3 DES OE of *Parietichytrium* sp. SEK358 in fed-batch culture. DCW and TFA of fed-batch culture (1 L jar fermenter) of **c** C20ELO KO/ω3DES OE and **d** Δ4DES KO/ω3 DES OE. The glucose concentration, DO level, and pH of the medium were monitored and maintained, as described in the "Methods" section. **e** Production of EPA using C20ELO KO and C20ELO KO/ω3DES OE in fed-batch culture. **f** Production of n-3DPA using Δ4DES KO and Δ4DES KO/ω3DES OE in fed-batch culture. C20ELO KO and C20ELO KO/ω3DES OE strains for EPA production, and Δ4DES KO and Δ4DES KO/ω3DES OE strains for n-3DPA production used in fed batch culture were derived from *Parietichytrium* sp. SEK364 and SEK358, respectively. Because SEK364 and SEK358 mutants showed superior production of EPA and n-3DPA, respectively, in fed batch culture under the conditions used. The LC-PUFA profiles of both mutant strains are very similar (Supplementary Fig. S12a, b). All data shown are the mean ± SD. ($n = 3$). The $n$ values are numbers of replicates.

PUFAs compared with WT (Fig. 3b). The levels of DCW and TFA in the C20ELO KO/ω3DES OE mutant continuously increased during the culture period to reach approximately 144 g/L and 76 g/L, respectively, after 214 h in batch culture (Fig. 3c).

Similarly, approximately 84 g/L DCW and 43 g/L TFA were obtained after 190 h when the Δ4DES KO/ω3DES OE mutant was used for fed-batch culture (Fig. 3d). The yield of EPA and n-3DPA was increased by ω3DES overexpression, and the yield of

EPA reached approximately 11.4 g/L (15% of TFA) after 214 h using the C20ELO KO/ω3DES OE mutant (Fig. 3e) and that of n-3DPA reached approximately 4.8 g/L (11% of TFA) after 190 h using Δ4DES KO/ω3DES OE mutant (Fig. 3f). The yield of EPA and n-3DPA in *Parietichytrium* sp. mutants markedly increased in fed-batch culture compared with flask culture due to the elimination of two negative factors (glucose depletion and insufficient air supply) in flask culture. Collectively, the use of gene manipulation and fed-batch culture in this study made *Parietichytrium* sp. suitable for mass production of EPA and n-3DPA.

## Discussion

Thraustochytrids have been characterized by anaerobic biosynthesis of DHA via PUFA-S, which is considered to be obtained from marine bacteria by horizontal gene transfer[4,11]. In contrast to the previously known thraustochytrids, *Parietichytrium* sp. has no PUFA-S gene and synthesizes DHA solely via the ELO/DES pathway (type III LC-PUFA synthesis system), as shown in this study. The distribution of type III LC-PUFA synthesis systems in thraustochytrids, and the relationship of thraustochytrid species possessing different LC-PUFA synthesis systems remain unknown. However, we consider that the genus *Parietichytrium* has the characteristic of having a type III LC-PUFA synthesis pathway, because all strains of *Parietichytrium* sp. tested (Supplementary Fig. S12c) showed the characteristic GC profile of type III pathway, as shown in Fig. 1b.

LC-PUFA synthesis pathways similar to type III have been found in molds, such as *Mortierella alpina*[36], and photosynthetic haptophytes, such as *Pavlova lutheri*[37]. However, *M. alpina* does not synthesize DHA due to the lack of Δ4DES, and *P. lutheri* primarily synthesizes EPA rather than DHA. Thus, the *Parietichytrium* sp. type III pathway can be thought of as a prototype of an aerobic DHA-synthesizing system. The evolutionary implications of the existence of three LC-PUFA synthesis systems among thraustochytrid species (Fig. 1f) are currently unknown. However, it is of interest that the PUFA-S system, which yields a simple LC-PUFA composition (Fig. 1b), disappeared, and the ELO/DES pathway responsible for the molecular diversity of LC-PUFAs was selected during evolution. In mammals, the diversity of LC-PUFAs plays a role in the diversity of membrane lipids, which are involved in membrane fluidity and functions[38].

In this study, we developed a new promoter, Smp1P, derived from SmDNAV, which is a double-stranded DNA virus capable of infecting thraustochytrids[34]. Smp1P is a promoter of the immediate-early gene of SmDNAV. The immediate-early gene promoter of double-stranded DNA viruses is often used in eukaryotic expression systems as a promoter with high transcription activity. The relative expression level of a marker protein (NeoR) under Smp1P was much higher than that under the ubiquitin promoter used in a previous study[35,39] (Supplementary Fig. S6b, c). Smp1P can be used for gene expression in not only *Parietichytrium* sp. but also other species of thraustochytrids, and thus will aid in the construction of highly efficient transformation systems for a wide range of thraustochytrids.

The yields of EPA and n-3DPA obtained via microbial production reported thus far are listed in Supplementary Table S1. Some marine microalgae are known to produce considerable amounts of EPA[40]. When the filamentous fungus *M. alpina* was cultured under low-temperature conditions, high expression of the ω3DES gene was induced, and the EPA content reached 26.4% of the TFA content and 1.8 g/L of culture fluid[41]. Recently, Wang et al. reported that *Aurantiochytrium* sp. transformed with a bacterial EPA-producing PUFA-S gene cluster produced 2.7 g/L EPA[42]. To date, the highest yield of EPA was achieved by

engineered *Yarrowia lipolytica*, which was transformed with several heterologous genes that constitute the ELO/DES pathway required for EPA synthesis[43]. The engineered oil yeasts eventually produced EPA at 55.6% of the TFA content, 120.7 mg/g of DCW and 5.5 g/L of culture fluid, by inactivating the PEX10 gene, which is involved in peroxisomal β-oxidation. Compared with the engineered *Y. lipolytica*, the *Parietichytrium* sp. mutant (C20ELO KO/ω3DES OE) had a lower EPA proportion in terms of TFA and DCW (15.6% of the TFA content and 78.9 mg/g of DCW); however, the mutant exhibited higher production of EPA per liter of culture (approximately 11.4 g/L), exhibiting the highest level of microbial production of EPA to date via a different approach from *Y. lipolytica* engineering. Compared to the engineered *Y. lipolytica* oils, the *Parietichytrium* oils contain a significant amount of n-6PUFAs, so further work may be needed to make the *Parietichytrium* oils more useful for human and fish supplements.

The physiological function of n-3DPA is not fully understood, although this LC-PUFA is present in the plasma, brain, retina, and heart of mammals[30]. Recent studies have reported that n-3DPA is converted to novel n-3 immunoresolvents in vivo and exhibits numerous physiological functions, such as anti-inflammatory and proresolving activities[29,44,45]. To date, there have been few reports on the details of microbial production of n-3DPA, other than reports of certain diatoms, thraustochytrids or *Y. lipolytica* employing designed myxobacterial PUFA-S[46] (Supplementary Table S1). We achieved the production of n-3DPA at 11% of the TFA content, 57 mg/g of DCW, and 4.8 g/L of culture using the mutant *Parietichytrium* sp. (Δ4DES KO/ω3DES OE). To the best of our knowledge, this is the highest yield of n-3DPA obtained by microbial fermentation. Mass production of n-3DPA enables the supply of pure n-3DPA, which will help to clarify the physiological and pharmacological properties of n-3DPA. Furthermore, *Parietichytrium* sp., which naturally possesses all genes required for DHA synthesis in its genome, will enable the production of desired LC-PUFAs other than EPA and n-3DPA by disrupting the target gene in the ELO/DES pathway.

## Methods

**Thraustochytrid strains.** *T. aureum* ATCC 34304 was obtained from the American Type Culture Collection (USA). *Parietichytrium* sp. SEK358, SEK364[47], I65-24A, and *A. limacinum* mh0186[48] were isolated from seawater from the Ishigaki Islands, Iriomote Islands, Ishigaki Islands, and Yaeyama Islands, Okinawa, Japan, respectively. *Parietichytrium* sp. SEK364 was identified as *Parietichytrium sarkarianum* by analyses of 18S rRNA gene, morphology and cultural properties[47]. The *Parietichytrium* strains used in each experiment are indicated in the legend of each Figure and Table.

**Culture of thraustochytrids.** Thraustochytrid strains were grown in 20 mL of GY medium [3% glucose and 1% yeast extract in 50% artificial sea water(3% NaCl, 0.07% KCl, 1.08% MgCl$_2$ 6H$_2$O, 0.54% MgSO$_4$ 7H$_2$O, 0.1% CaCl$_2$ 2H$_2$O)] with 0.1% vitamin mixture (0.2% vitamin B$_1$, 0.001% vitamin B$_2$, and 0.001% vitamin B$_{12}$) and 0.2% trace elements (3% EDTA di-sodium, 0.15% FeCl$_3$·6H$_2$O, 3.4% H$_3$BO$_3$, 0.43% MnCl$_2$·4H$_2$O, 0.13% ZnSO$_4$·7H$_2$O, 0.026% CoCl$_2$·6H$_2$O, 0.013% NiSO$_4$·6H$_2$O, 0.001% CuSO$_4$·5H$_2$O, and 0.0025% Na$_2$MoO$_4$·2H$_2$O) in a 100 mL flask at 25 °C with rotation at 150 rpm. Potato dextrose agar (PDA) plates (0.8% potato dextrose agar, 50% artificial sea water, 2% agar) containing appropriate antibiotics were used to select *Parietichytrium* sp. and *T. aureum* mutants. The growth of these strains was monitored by measuring the optical density at 600 nm (OD 600), dry cell weight (DCW), and consumption of glucose in the medium. The glucose concentration was measured by the Glucose CII test (FUJIFILM Wako Pure Chemical Co., Japan) and the concentration of dissolved oxygen (DO) was measured by a DO meter.

**GC analysis to determine the fatty acid composition.** Cells were cultured in a 100 mL flask containing 40 mL of GY medium at 25 °C with shaking at 150 rpm. Two mL of culture fluid was harvested, and cells were collected by centrifugation (2500 × *g*, 3 min), dried by lyophilization, and their DCW was measured. The fatty acids were extracted as fatty acid methyl esters (FAMEs) from dried cells as described in ref.[49]. FAMEs were analyzed by GC-2014 (Shimadzu) equipped with an ULBON HR SS-10 column (Shinwa Chemical Industries). The column

temperature was programmed to increase from 160 to 200 °C by 2 °C/min, and then from 200 to 220 °C by 5 °C/min.

**Genome sequencing and assembly**. Genomic DNA from thraustochytrids was extracted by the standard phenol–chloroform method. The draft genome sequences of *T. aureum* ATCC 34304 and *Parietichytrium* sp. I65-24A were obtained using a 454 GS FLX system (Roche Diagnostics) and assembled using Newbler assembler version 2.7 software (Roche Diagnostics). The genomic DNA from *T. aureum* was sequenced using a paired-end sequencing strategy, and a total of 369 Mbp was obtained from 1,046,218 reads (approximately 9-fold coverage). Assembly of all reads resulted in 14,205 contigs (>500 bp) and 463 scaffolds. The genomic DNA from *Parietichytrium* sp. was sequenced using a whole-genome shotgun and a paired-end sequencing strategy, and a total of 1556 Mbp was obtained from 4,511,709 reads (approximately 29-fold coverage). Assembly of all reads resulted in 11,544 contigs (>500 bp) and 360 scaffolds. The draft genome sequences of *Parietichytrium* sp. and *T. aureum* have been deposited into DDBJ under accession numbers BLSF01000001 to BLSF01000360 and BLSG01000001 to BLSG01000463, respectively.

**Expression of DES and ELO genes in *S. cerevisiae***. The candidate genes for ELO and DES were extracted from the *Parietichytrium* draft genome database by local BLAST using previously known ELO/DES genes as query sequences. The ORFs of the putative ELO (ELO-1, ELO-2, and ELO-3) and DES (DES-1, DES-2, DES-3, DES-4, DES-5, and DES-6) were amplified by PCR using cDNA or genomic DNA of *Parietichytrium*, and inserted into the MCS of pYES2/CT (Invitrogen). The ω3DES of *S. diclina*[33] and *M. alpina*[50] were obtained from genomic DNA and cDNA, respectively, by PCR, and inserted into pYES2/CT. The expression vectors were introduced into *S. cerevisiae* INVSc1 (Invitrogen) using the lithium acetate method[51]. The transformants were selected by plating on synthetic agar plates lacking uracil (SC-ura). *S. cerevisiae* transformants harboring ELO and DES were cultured in SC-ura medium containing 2% glucose at 25 °C for 1 day, and then cultured for an additional 1 day in SC-ura medium containing 2% galactose and 50 μM fatty acids (substrate), as described in Supplementary Fig. S1a. The cells were collected by centrifugation at $2000 \times g$ for 10 min and lyophilized. The fatty acid profiles were obtained by GC analysis. ELO or DES activity was expressed as follows: activity (%) = product area × 100/(substrate area + product area).

**Quantitative real-time PCR**. Real-time PCR was performed using an Mx3000P qPCR System (Agilent Technologies) with TB Green Premix Ex Taq II (Tli RNaseH Plus) (Takara-Bio) using cDNA from *Parietichytrium* (2 days culture) as a template. The copy numbers of three ELO and six DES were measured using plasmids containing the ORF of each gene as a standard.

**Disruption of genes in *Parietichytrium* sp. and *T. aureum***. The genes of Δ4DES and C20ELO in *Parietichytrium* sp., and Δ4DES and PUFA-S in *T. aureum* were disrupted by homologous recombination using targeting constructs (Supplementary Fig. S13) as described in the previous report[35]. The linearized knockout targeting constructs containing antibiotic resistance gene were obtained by PCR, introduced into thraustochytrids with a PDS-1000/He (BioRad) using 1550 psi (for *Parietichytrium* sp.) or 1100 psi (for *T. aureum*) rupture disks, and then cultured on PDA plate containing appropriate antibiotics. After transfection, Δ4DES and C20ELO KO mutants of *Parietichytrium* sp. SEK358 (Supplementary Fig. S13a, b) were selected on PDA medium containing 0.5 mg/mL of G418. Two different selection markers were used for disruption of the target genes of *Parietichytrium* sp. SEK364 and *T. aureum* (Supplementary Fig. S13c–e). After transfection of C20ELO knockout construct containing NeoR (Supplementary Fig. S13c), *Parietichytrium* sp. SEK364 was cultured on 2 mg/mL G418, then first allele knockout strain was selected by PCR. For the disruption of second allele of C20ELO of *Parietichytrium* sp. SEK364, hygromycin resistance gene (HygR) containing knockout construct (Supplementary Fig. S13c) was introduced into first allele knockout strain, then cultured on 2 mg/mL hygromycin containing PDA. The first allele of Δ4DES in *T. aureum* was replaced with the knockout construct containing the Blasticidin S resistance gene (BlaR) expression cassette (first allele KO construct) and selected on PDA medium containing 0.2 mg/mL of Blasticidin S (Supplementary Fig. S13d). The second allele was replaced with the construct containing the GFP-fused zeocin resistance gene (ZeoR) expression cassette (second allele KO construct) and selected on PDA medium containing 20 mg/mL of zeocin (Supplementary Fig. S13d). In addition to zeocin resistance, fluorescence derived from GFP-ZeoR was used as indicator to select transfectants because of low antibiotic efficiency of zeocin. The first allele of PUFA-S in *T. aureum* was replaced with the disruption construct containing NeoR expression cassette (first allele KO construct, Supplementary Fig. S13e) and selected on PDA medium containing 2 mg/mL of G418. The second allele was replaced with the construct containing the hygromycin resistance gene (HygR) expression cassette (second allele KO construct, Supplementary Fig. S13e) and selected on PDA medium containing 2 mg/mL of hygromycin, generating the PUFA-S KO strain. The PUFA-S-disrupted and Δ4DES-disrupted mutant (PUFA-S/Δ4DES KO) was obtained by introducing the Δ4DES knockout construct into the PUFA-S KO strain. The mutants were selected by

PCR, and the gene disruption of mutants was confirmed by Southern blot analysis using WT DNA as a control.

**Development of a high-expression promoter for *Parietichytrium* sp**. To overexpress ω3DES, which is a bottleneck for EPA or n-3DPA production by *Parietichytrium*, we developed high-expression promoters in this study. SmDNAV is a double-stranded DNA virus infecting *S. minutum*[34]. To identify the immediate early genes of SmDNAV, DNA microarray was performed in SmDNAV-infected *S. minutum* treated with or without cycloheximide. The 1 kbp upstream sequence of immediate early genes was isolated from the genomic DNA of SmDNAV and inserted upstream of NeoR. Then, the promoter activity was assessed by the transformation efficiency or mRNA expression level of NeoR, which was quantified by real-time PCR. The promoter region demonstrating the highest transcriptional activity was used as the SmDNAV-derived promoter (Smp1P) in this study.

**Generation of ω3DES-overexpressing mutants**. The codon optimized ω3DES gene of *S. diclina* was chemically synthesized (Biomatik), and inserted between the Smp1 or EF1α promoter and ubiquitin terminator (Supplementary Fig. S6d). The linearized expression construct was prepared by PCR using the plasmid as a template, and then introduced into Δ4DES KO and C20ELO KO strains of *Parietichytrium* sp. by biolistic transformation with a PDS-1000/He (BioRad). C20ELO KO/ω3DES OE was selected by 1 mg/mL of Blasticidin-containing PDA. Δ4DES KO/ω3DES OE was selected by 1 mg/mL hygromycin-containing PDA.

**Tracing the metabolism of stable isotope-labeled fatty acids**. Deuterium-labeled palmitic acid (d31-C16:0) and $^{13}C_{18}$-labeled oleic acid ($^{13}C_{18}$-C18:1) were purchased from Cambridge Isotope Laboratories, Inc. $^{13}C_{18}$-C18:1 was added to the medium at a final concentration of 0.25 mM. Cells were collected 4, 8, 24, and 48 h after adding stable isotope-labeled fatty acids. The cell pellet and supernatant were separated by centrifugation at $5,000 \times g$ for 3 min. The cell pellet was dissolved in 200 μL of distilled water. Then, the pellet was crushed at 3000 rpm for 60 s using a bead beater (μT-12, TAITEC) with glass beads (diameter: 0.4 mm, AS ONE Corp.) and kept on ice for 60 s. This procedure was repeated three times to prepare the cell lysate. Cellular lipids were extracted from 90 μL of cell lysate by adding 375 μL of chloroform/methanol (1:2, v/v) containing 50 μM internal standard (C12:0). For saponification, 10 μL of 4 M KOH was added. After incubation at 37 °C for 3 h, 10 μL of 2 M acetic acid, 125 μL of chloroform, and 125 μL of water were added, and the organic phase was separated by centrifugation at $17,500 \times g$ for 3 min. Then, 150 μL of the organic phase was transferred to vials, and the stable isotope-labeled fatty acids were measured using LC-ESI MS/MS (3200 QTRAP, SCIEX) equipped with an InertSustain C18 column (2.1 × 150 mm, 5 μm, GL Sciences) using a gradient starting with 10% solvent B (methanol with 2.5 mM ammonium acetate) in solvent A (distilled water with 2.5 mM ammonium acetate) and reaching 90% solvent B for 1 min, followed by 95% solvent B for 15 min. The column was equilibrated for 5 min before the next run. Isotope-labeled fatty acids were detected by MRM or pseudo-MRM, for which the conditions are described in Supplementary data set 1.

**LC-ESI MS/MS to determine the lipid composition**. The cell pellet and supernatant were separated by centrifugation at $5000 \times g$ for 3 min, and the DCW was measured after the pellet was lyophilized. The 2 mg of dried cells were dissolved in 300 μL of chloroform/methanol (2:1, v/v) containing 10 μM PC22:0 (11:0/11:0), 10 μM LPC13:0, and 20 μM TG36:0 (12:0/12:0/12:0) as internal standards, and then crushed by sonication for 60 s and kept on ice for 60 s. This procedure was repeated three times. After incubation at 25 °C for 30 min, the supernatant and cell debris were separated by centrifugation at $17,500 \times g$ for 3 min. The $240 \times g$ of supernatant was mixed with 60 μL of water, and then centrifuged at $17,500 \times g$ for 3 min. Forty microliters of the organic phase was transferred into autoinjector vials containing 960 μL of 2-propanol, and then the cellular lipids were measured by LC-ESI MS/MS using a high-performance liquid chromatography system (1200 series, Agilent Technologies) equipped with a mass spectrometer (3200 QTRAP LC-MS/MS system). A binary solvent gradient with a flow rate of 200 μL/min was used to separate phospholipids and neutral lipids by reversed-phase chromatography using an InertSustain C18 column (2.1 × 150 mm, 5 μm)[39,52]. The gradient was started with 0% solvent B1 (2-propanol with 0.1% formic acid and 0.028% ammonia) in solvent A1 (acetonitrile/methanol/water, 19:19:2, v/v/v containing 0.1% formic acid and 0.028% ammonia) and was maintained for 3 min. The gradient reached 40% B for 24 min, then 50% B for 26 min, and was maintained for 3 min. The gradient was returned to the starting conditions for 1 min and the column was equilibrated for 7 min before the next run. LPC, PC, LPE, PE, diacylglycerol (DAG), and TAG containing C16:0, C18:0, C18:1, C18:2, C18:3, C20:3, C20:4, C20:5, C22:4, C22:5, and C22:6 fatty acids were detected using MRM. The peak intensity of each glycerolipid species was analyzed using MultiQuant 3.0.1 (SCIEX). MRM conditions are described in Supplementary data set 1.

**Fed-batch culture of *Parietichytrium* sp**. Mutant strains of *Parietichytrium* sp. SEK364 (C20ELO KO and C20ELO KO/ω3DES OE) *Parietichytrium* sp. SEK358 (Δ4DES KO and Δ4DES KO/ω3DES OE) were precultured in 500-mL flasks containing 100 mL of GY medium at 28 °C with shaking at 120 rpm. After glucose consumption, the cells were transferred to a Small Scale Bioreactor BMP-Z (ABLE

Co. & Biott Co., Ltd) equipped with a 1 L vessel containing 400 mL of GY medium, the temperature and pH of which were maintained at 28 °C and 6.8 ± 0.2, respectively. The DO level in the medium was maintained above 10% by adjusting the agitation speed (from 600 to 1200 rpm) and air flow rate (from 0.25 to 0.5 L/min). The glucose concentration was continuously monitored and maintained below 60 g/L throughout cultivation by adding feed solution (50% glucose). Samples were collected at the time points indicated in Fig. 3, and the DCW, TFA, EPA, and n-3DPA levels were measured by the method described in "Methods" section.

**Statistics and reproducibility.** The results are expressed as mean ± SD.

**Reporting summary.** Further information on research design is available in the Nature Research Reporting Summary linked to this article.

## Data availability
The authors declare that all data supporting the findings of this study are available within the article and its supplementary information (Figs. S1–S13 and Table S1), supplementary data 1, 2 or from the corresponding author upon reasonable request. Accession number of genes encoding ELO-1, ELO-2, ELO-3, DES-1, DES-2, DES-3, DES-4, DES-5, DES-6, and Smp1 promoter are LC530191, LC530192, LC530193, LC530194, LC530195, LC530196, LC530197, LC530198, LC530199, and LC599980 in DDBJ databases, respectively. The draft genome sequences of *Parietichytrium* sp. and *Thraustochytrium aureum* have been deposited into DDBJ as accession numbers BLSF01000001 to BLSF01000360 and BLSG01000001 to BLSG01000463, respectively.

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

## Acknowledgements

This work was supported by The Science and Technology Research Promotion program for Agriculture, Fisheries and Food Industry, Japan (26050A). We acknowledge E. Abe, S. Kim (Kyushu University), N. Nagano, Y. Taoka, A. Matsuda (University of Miyazaki), M. Ueda and Y. Takeuchi (Konan University) for their technical assistance and valuable suggestions.

## Author contributions

Conceptualization: M.I., R.H. and K.S.; Funding acquisition: M.I.; Project administration, M.I. and Y.O.; Investigation, Y.I., H.G., R.H., K.S., T.S., Yu.I., S.M., I.S., M.T., M.K·, K.T. and N.O.; Resources, D.H., M.H. and Y.T.; Validation: M.I., Y.I. and H.G.; Visualization, H.G., Y.I., T.S. and N.O.; Writing: M.I., Y.I. and H.G.

## Competing interests

The authors declare no competing interests.
