## [Transparent Peer Review File · Communications Biology]

Reviewers' comments:

Reviewer #1 (Remarks to the Author):

The manuscript submitted by Ishibashi, Goto, Hamaguchi, Sakaguchi et al. describes the thorough characterization of the biosynthetic pathway of LC-PUFAs in thraustochytrids using an excellent molecular biological approach. This study not only provides a basic understanding of the biosynthetic pathway of LC-PUFAs, but also examines the practical production of useful LC-PUFAs based on the results obtained in this study.

This is a carefully done study and findings are of considerable interest. You have submitted quite a fascinating manuscript and I sure would like to see it printed soon. Although I have no serious criticisms regarding methodology, results, and interpretation of results, I have a few comments.

1. Legends to Figures, Figure 3: You mentioned that "C20ELO KO and C20ELO KO/ ω 3DES OE strains for EPA production, and D4DES KO and D4DES KO/ ω 3DES OE strains for n-3DPA production used in fed batch culture were derived from *Parietichytrium* sp. SEK364 and SEK358, respectively". Why did you use different strains of *Parietichytrium* sp. to produce the mutant strains?

2. Fig. 1F: It is very interesting to note that the biosynthetic pathways of LC-PUFAs in thraustochytrids can be divided into three groups. You mentioned that "the evolutionary implications of the existence of three DHA synthesis systems among thraustochytrid species (Fig. 1F) are currently unknown", but what do you think about the distribution of these three groups within the thraustochytrid species? Are there any characteristics?

3. Supple. Figs. S5 and S6: Why is the amount of PC36:5 or PC38:5 per DCW much higher than that of TG54:5 or TG56:6 per DCW although there are other molecular species of PC and TG including 20:5 and 22:5? In thraustochytrids, the amount of TG is usually higher than that of phospholipids.

Minor revisions are listed below.

4. Abstract: There are some abbreviations that I think need to be explained like C20ELO, Δ 4DES, Δ 3DES, and n-3DPA.

5. Introduction, 1st paragraph: "Docosahexaenoic acid (DHA, C22:6)" should be "Docosahexaenoic acid (DHA, C22:6n-3)".

6. Introduction, 3rd paragraph: "n-3DPA" should be "n-3 docosapentaenoic acid (n-3DPA, C22:5n-3)".

7. Results, 1st paragraph: "n-6 docosapentaenoic acid (n-6DPA, C22:5n-3)" should be "n-6 docosapentaenoic acid (n-6DPA, C22:5n-6)".

8. Results, 5th paragraph: "the Δ 9DES homolog does not is consistent with the results obtained in this study" should be "the Δ 9DES homolog does not consistent with the results obtained in this study".

9. References: Please align the style of References correctly. For example, in the title of the paper, only the word at the beginning of the title is capitalized in some cases and not in others. And scientific names, like *Saccharomyces cerevisiae*, should be written in italics. Furthermore, there is a mixture of abbreviated and unabbreviated journal names.

10. Fig 2B: After the 3 day of cultivation time, the \circ (open circle, light blue) and Δ (open triangle, red) symbols should disappear.

11. I am concerned about the mixture of *Parietichytrium* sp. SEK364 and *Parietichytrium sarkarianum* in the manuscript. Are they being used differently?

Reviewer #2 (Remarks to the Author):

I find this an interesting paper describing much work, which has been well carried out using appropriate methodology. In particular, *Parietichytrium* was shown convincingly to use a des/elo pathway rather than PUFA-S as in other *thraustochytrium* spp.

The following detailed points are made for the author's attention:

A general comment that should be addressed is that the final oils produced by *Parietichytrium* all contain significant amounts of n-6PUFAs. Since the ratio of n-3/n-6 PUFA is important for nutrition (in humans as well as fish), further work is probably needed to make *Parietichytrium* oils more useful. Page 2, para 3, line 3. While it is true that dietary LC-PUFAs come from fish, it is algae that produce them. Re-word?

Page 2, para.3,last sentence. *Thraustochytrids* are already used successfully. Re-word?

Page 2, last line. The correct name is triacylglycerol (abbrev. TAG). Please use this throughout the manuscript.

Page 3,para.3. Some re-wording is needed. Three species are named at first but in line 5, 'two' are mentioned.

Page 3. A figure or, better, a table which shows the complete fatty acid composition should be included in the main body of the manuscript. At present, the constant reference to LC-PUFA composition is often of limited use. This applies to later discussion of pathways(e.g.page 5). So it is IMPORTANT to include a table of total fatty acids.

Page 3, para.4, line 4. PAM is an odd abbreviation for 16:0. Surely PA would be better, if it has to be used?

Page 6, para.3, line 1. Better to have a list of abbreviations used

Page 7, para.3. I note the usefulness of Smp1P. This is a good discovery by the authors.

Page 7, para.4. Actually some marine algae already produce considerable EPA (over 20%). Please re-word.

Page 8,top of para.2. This is wrong and needs correcting. For example, brain contains 70-times as much DHA as DPA.

Legends to figures. For the n value given, is this for independent biological samples or for replicates? Please clarify.

Thank you for your email dated Aug 2, 2021, together with the comments from the reviewers with regard to our manuscript (COMMSBIO-21-1721-T). According to the reviewers' suggestions, we revised the manuscript.

Reviewers' comments:

All corrections are shown in red text in the revised manuscript.

Reviewer #1 (Remarks to the Author):

The manuscript submitted by Ishibashi, Goda, Hamaguchi, Sakaguchi et al. describes the thorough characterization of the biosynthetic pathway of LC-PUFAs in thraustochytrids using an excellent molecular biological approach. This study not only provides a basic understanding of the biosynthetic pathway of LC-PUFAs, but also examines the practical production of useful LC-PUFAs based on the results obtained in this study.

This is a carefully done study and findings are of considerable interest. You have submitted quite a fascinating manuscript and I sure would like to see it printed soon. Although I have no serious criticisms regarding methodology, results, and interpretation of results, I have a few comments.

1. Legends to Figures, Figure 3: You mentioned that “C20ELO KO and C20ELO KO/ ω 3DES OE strains for EPA production, and D4DES KO and D4DES KO/ ω 3DES OE strains for n-3DPA production used in fed batch culture were derived from *Parietichytrium* sp. SEK364 and SEK358, respectively”. Why did you use different strains of *Parietichytrium* sp. to produce the mutant strains?

We generated mutant strains from two different *Parietichytrium* sp. (strains SEK358 and SEK364). The profiles of LC-PUFAs of the two *Parietichytrium* strains and their mutants are almost the same, as shown in Supplemental Fig. S12A, B. As a result of preliminary experiments of fed-batch cultures using mutant strains, we found that SEK364 and SEK358 mutants showed superior productions of EPA and n-3DPA, respectively. The reason remains unknown at present, but we provided data using SEK364 mutants for EPA production (Fig. 3C, E) and those using SEK358 mutants for n-3DPA production (Fig. 3D, F). We added Supplemental Fig. S12A, B and the statements to the legend of Fig. 3 in the revised manuscript (p16, lines 14-16).

2. Fig. 1F: It is very interesting to note that the biosynthetic pathways of LC-PUFAs in thraustochytrids can be divided into three groups. You mentioned that “the evolutionary implications of the existence of three DHA synthesis systems among thraustochytrid species (Fig. 1F) are currently unknown”, but what do you think about the distribution of these three groups within the thraustochytrid species? Are there any characteristics?

The distribution of type III LC-PUFA synthesis systems in thraustochytrids, and the relationship of thraustochytrid species possessing different LC-PUFA synthesis systems remain unknown. However, we consider that the genus *Parietichytrium* has the characteristic of having a type III LC-PUFA synthesis pathway, because all strains of *Parietichytrium* sp. tested (Supplemental Fig. S12C) showed the characteristic GC profile of type III pathway, as shown in Fig. 1B. This statement (p 8, lines 22-27) and Fig. S12C were added to the revised manuscript.

3. Supple. Figs. S5 and S6: Why is the amount of PC36:5 or PC38:5 per DCW much higher than that of TG54:5 or TG56:6 per DCW although There are other molecular species of PC and TG including 20:5 and 22:5.? In thraustochytrids, the amount of TG is usually higher than that of phospholipids.

We agree with the reviewer's comments that the amounts of TGs are usually higher than those of phospholipids in thraustochytrids. The values on the vertical axis of Supplementary Fig. S5 and S6 (Supplementary Fig. S7 and S8 in the revised manuscript) are based on the peak intensities of LC-MS. The peak intensities of phospholipids are much higher than those of TGs, as shown in Supplementary Fig. S9 under the conditions used. Thus, we need standards to quantify PC and TG with different fatty acid molecules by LC-MS; however, unfortunately, it is difficult to obtain standards for each PC and TG molecule. We added Supplemental Fig. S9 and these statements to the legends of Fig. S7 and S8 in the revised manuscript.

We modified Supplementary Fig. S7 and Fig. S8. Fig. S7(A), (D) show all PC and TG molecules containing C20:5, C20:4, respectively, and Fig. S8(A), (D) show all PC and TG molecules containing C22:4, C22:5, respectively, to address the queries from the reviewer.

Minor revisions are listed below.

4. Abstract: There are some abbreviations that I think need to be explained like C20ELO, Δ 4DES, Δ 3DES, and n-3DPA.

We added the full spelling and description of abbreviations when first used. We also added a list to summarize all abbreviations (p2), as suggested by reviewer 2.

5. Introduction, 1st paragraph: "Docosahexaenoic acid (DHA, C22:6)" should be "Docosahexaenoic acid (DHA, C22:6n-3)".

Corrected (p 3, line 17)

6. Introduction, 3rd paragraph: "n-3DPA" should be "n-3 docosapentaenoic acid (n-3DPA, C22:5n-3)".

Corrected (p 4, line 3).

7. Results, 1st paragraph: "n-6 docosapentaenoic acid (n-6DPA, C22:5n-3)" should be "n-6 docosapentaenoic acid (n-6DPA, C22:5n-6)".

Corrected (p 4, lines 21-22).

8. Results, 5th paragraph: "the Δ 9DES homolog does not is consistent with the results obtained in this study" should be "the Δ 9DES homolog does not consistent with the results obtained in this study".

Corrected (p 6, line 11).

9. References: Please align the style of References correctly. For example, in the title of the paper, only the word at the beginning of the title is capitalized in some cases and not in others.

And scientific names, like *Saccharomyces cerevisiae*, should be written in italics. Furthermore, there is a mixture of abbreviated and unabbreviated journal names.

We revised the style of references.

10. Fig 2B: After the 3 day of cultivation time, the ○ (open circle, light blue) and △ (open triangle, red) symbols should disappear.

We revised Fig. 2B, as suggested.

11. I am concerned about the mixture of *Parietichytrium* sp. SEK364 and *Parietichytrium sarkarianum* in the manuscript. Are they being used differently?

SEK364 was identified as *Parietichytrium sarkarianum*, and, thus, they have the same meaning.

Reviewer #2 (Remarks to the Author):

I find this an interesting paper describing much work, which has been well carried out using appropriate methodology. In particular, *Parietichytrium* was shown convincingly to use a des/elo pathway rather than PUFA-S as in other *thraustochytrium* spp.

The following detailed points are made for the author's attention:

A general comment that should be addressed is that the final oils produced by *Parietichytrium* all contain significant amounts of n-6PUFAs. Since the ratio of n-3/n-6 PUFA is important for nutrition (in humans as well as fish), further work is probably needed to make *Parietichytrium* oils more useful.

We agree with the reviewer's comment. Compared with the engineered *Y. lipolytica* oils, *Parietichytrium* oils contain a significant amount of n-6PUFAs, so further work may be needed to make the *Parietichytrium* oils more useful for human and fish supplements. This comment was added to the text of the revised manuscript (p 9, lines 23-26).

Page 2, para 3, line 3. While it is true that dietary LC-PUFAs come from fish, it is algae that produce them. Re-word?

We added the sentence (p 3, lines 32-33).

Page 2, para.3,last sentence. *Thraustochytrids* are already used successfully. Re-word?

We modified the sentence (p 3, line 41).

Page 2, last line. The correct name is triacylglycerol (abbrev. TAG). Please use this throughout the manuscript.

We corrected throughout the revised manuscript.

Page 3,para.3. Some re-wording is needed. Three species are named at first but in line 5, 'two' are mentioned.

Corrected (p 4, lines 19-20).

Page 3. A figure or, better, a table which shows the complete fatty acid composition should be included in the main body of the manuscript. At present, the constant reference to LC-PUFA composition is often of limited use. This applies to later discussion of pathways (e.g.page 5). So it is IMPORTANT to include a table of total fatty acids.

We added the total fatty acid compositions of three thraustochytrids (Table 1) and the statement was added in the revised manuscript (p 4, lines 25-28).

Page 3, para.4, line 4. PAM is an odd abbreviation for 16:0. Surely PA would be better, if it has to be used?

We used PA as an abbreviation of palmitic acid in the revised manuscript.

Page 6, para.3, line 1. Better to have a list of abbreviations used

We added a list of abbreviations in page 2, as suggested by the reviewer.

Page 7, para.3. I note the usefulness of Smp1P. This is a good discovery by the authors.

Thank you for the indication.

Page 7, para.4. Actually some marine algae already produce considerable EPA (over 20%). Please re-word.

We modified the sentence (p 9, lines 8-9).

Page 8, top of para.2. This is wrong and needs correcting. For example, brain contains 70-times as much DHA as DPA.

We modified the sentence (p 9, lines 28-29).

Legends to figures. For the n value given, is this for independent biological samples or for replicates? Please clarify.

The n values are numbers of replicates. This statement was added to the legends of all figures to be required.

REVIEWERS' COMMENTS:

Reviewer #1 (Remarks to the Author):

I thank the authors for their thoughtful responses to my comments.

I agree with the authors' corrections and comments. The revised manuscript looks almost fine, however, I found several things that need to be corrected on the revised manuscript. I hope these are helpful.

1. Abbreviations, Line 9: "Phosphatidylcholine" is probably more common.
2. Abbreviations, Line 11: "Sicyoidochytrium minutum" should be italic.
3. Page 5, line 17: "wild type (WT)" should be "wildtype (WT)"
4. Page 5, line 21: "wildtype (WT)" should be "WT".
5. Page 5, line 21: "13C-labeled oleic acid" should be "13C-labeled OA".
6. References: #6, #25, and #42 should be aligned correctly in format. For #9, "Delta12-fatty acid desaturase" should be " Δ 12-fatty acid desaturase".
7. Table 1: I think the table is incomplete. There should be a "line" at the top and at the bottom. (n-6 DPA) and (n-3 DPA) would be better as (n-6DPA) and (n-3DPA).
8. Legends to Figures, lines 11-12: I think "C, LC-PUFA profiles of WT and Δ 4DES KO of *Parietichytrium* sp. SEK358" would be better.
9. Supplemental Fig. S1B: C20:2 should be C20:2n-6.

Reviewer #2 (Remarks to the Author):

The authors have revised their m/s satisfactorily and have taken account of all of the reviewers' comments.

I have one minor comment only:

On page 4, lines 26-28 a new sentence is inserted. However, LA in *Parietichytrium* is stated to be 'high'. 3.1% cannot be considered high. Please re-word.

Reviewer #1 (Remarks to the Author):

I thank the authors for their thoughtful responses to my comments.

I agree with the authors' corrections and comments. The revised manuscript looks almost fine, however, I found several things that need to be corrected on the revised manuscript. I hope these are helpful.

1. Abbreviations, Line 9: "Phosphatidylcholine" is probably more common.

Corrected. P 1, line 41.

2. Abbreviations, Line 11: "Sicyoidochytrium minutum" should be italic.

Corrected. P 1, line 42.

3. Page 4, line 17: "wild type (WT)" should be "wildtype (WT)"

Corrected. P 4, line 17.

4. Page 4, line 21: "wildtype (WT)" should be "WT".

Corrected. P 4, line 21.

5. Page 4, line 30: "13C-labeled oleic acid" should be "13C-labeled OA".

Corrected. P 4, line 30.

6. References: #6, #25, and #42 should be aligned correctly in format. For #9, "Delta12-fatty acid desaturase" should be "Δ12-fatty acid desaturase".

Corrected.

7. Table 1: I think the table is incomplete. There should be a "line" at the top and at the bottom. (n-6 DPA) and (n-3 DPA) would be better as (n-6DPA) and (n-3DPA).

Corrected.

8. Legends to Figures, lines 11-12: I think "C, LC-PUFA profiles of WT and Δ4DES KO of *Parietichytrium* sp. SEK358" would be better.

Corrected. P 18, lines 11-12.

9. Supplemental Fig. S1B: C20:2 should be C20:2n-6.

Corrected.

Reviewer #2 (Remarks to the Author):

The authors have revised their m/s satisfactorily and have taken account of all of the reviewers' comments.

I have one minor comment only:

On page 3, lines 26-28 a new sentence is inserted. However, LA in *Parietichytrium* is stated to be 'high'. 3.1% cannot be considered high. Please re-word.

Corrected. LA is eliminated. P 3, line 27.

1) Please ensure that you upload the attached checklist when submitting your revision as well as your completed editorial policy checklist and reporting summary documents.

OK

2) Please address the remaining minor point from the reviewer and please confirm that you have done in this your cover letter rather than highlighting changes.

Done.

3) Please move your 'data deposition' statement to just below the methods and please change the title to 'data availability'.

Done.

4) Your author 'contribution statement' and 'competing interests' should also come after the methods

Done.

5) Please remove titles from the bottom of your figures.

Done.

6) Please label figure panels with lower case letters and update corresponding text accordingly. This applies to main and supplementary figures.

Done.

7) Please add data points to all line graphs (please see attached table for guidance. This includes 1e, 2b-e, 2g-j, 3c-f.

Done.

8) Please move all methods details from supplementary to main. You do not need to worry about a word limit for methods.

Done.

9) Supplementary tables S1 and S3 as supplementary data instead. These should be in excel format and called 'supplementary data set'.

Done.

10) Please pay careful attention to formatting guidelines for tables (see attached).

Done.